# New Targets for Antiviral Therapy: Inhibitory Receptors and Immune Checkpoints on Myeloid Cells

**DOI:** 10.3390/v14061144

**Published:** 2022-05-25

**Authors:** Yanni Liu, Paul Nicklin, Yuan He

**Affiliations:** 1Center for Infectious Disease Research, School of Medicine, Tsinghua University, Beijing 100084, China; liuyn2020@mail.tsinghua.edu.cn; 2Tsinghua-Peking Center for Life Sciences, Tsinghua University, Beijing 100084, China; 3Research Beyond Borders, Boehringer Ingelheim, 88397 Biberach an der Riss, Germany; paul.nicklin@boehringer-ingelheim.com; 4Research Beyond Borders, Boehringer Ingelheim (China), Shanghai 200040, China

**Keywords:** inhibitory receptors, myeloid cells, immune responses, antiviral therapy

## Abstract

Immune homeostasis is achieved by balancing the activating and inhibitory signal transduction pathways mediated via cell surface receptors. Activation allows the host to mount an immune response to endogenous and exogenous antigens; suppressive modulation via inhibitory signaling protects the host from excessive inflammatory damage. The checkpoint regulation of myeloid cells during immune homeostasis raised their profile as important cellular targets for treating allergy, cancer and infectious disease. This review focuses on the structure and signaling of inhibitory receptors on myeloid cells, with particular attention placed on how the interplay between viruses and these receptors regulates antiviral immunity. The status of targeting inhibitory receptors on myeloid cells as a new therapeutic approach for antiviral treatment will be analyzed.

## 1. Introduction

The host immune system consists of a network of immune cell types working together in a coordinated fashion. Among them, the myeloid cells are increasingly understood to play an important role in detecting and combating viral infections as well as inducing antiviral immunity [1]. When myeloid cells recognize virus-infected cells or tissues, they quickly initiate innate immune responses by activating cellular pathways and release of cytokines [2,3,4,5]. Occasionally, the immune response becomes over-activated and fails to return to normal homeostasis, thereby leading to immunopathological damage [6,7,8,9,10]. Myeloid cells play a key role in maintaining immune homeostasis by orchestrating appropriate inflammatory responses [11]. They achieve this by expressing an array of receptors on the cell surface that integrate a multitude of activating and inhibitory signals [12].

There was a historical bias towards studying pathways activating immune cell function to fight against cancer and infections; however, there is emerging evidence that inhibitory signaling can regulate immune cell checkpoints, highlighting their potential as new therapeutic targets [13]. “Inhibitory receptor” was first termed by Lanier in 1998 [14]. The classical view of inhibitory receptors is that they contain immunoreceptor tyrosine-based inhibitory motifs (ITIMs) and inhibit activating pathways by interacting with paired activating receptors [12]. The function of inhibitory receptors on natural killer (NK) cells or T and B cells was reviewed recently [15,16,17]. There is now a growing interest in the role of inhibitory receptors on myeloid cells in sustaining an immunosuppressive tumor microenvironment and promoting immune evasion in cancer [18]. Consequently, myeloid cell-based checkpoint inhibitors are under development as cancer treatments and several are already undergoing clinical trials [18,19]. Evidence also indicates that inhibitory receptors on myeloid cells are involved in bacterial infection [20]. As for viral infections, the role of myeloid cells is important, multi-faceted and probably underestimated. To date, limited efforts are being made to identify new antiviral therapeutic targets on myeloid cells, especially the suppressive components. Recent studies reveal that viruses can also exploit inhibitory receptors on myeloid cells for immune evasion, leading to enhanced viral pathogenesis during acute and chronic infections [21]. The purpose of this review is to summarize the studies of inhibitory receptors on myeloid cells in relation to viral infections and to propose the development of myeloid cell immune checkpoint inhibitors as new antiviral therapies.

## 2. Structure of Inhibitory Receptors

Inhibitory receptors can be identified by ITIMs located in the cytoplasmic portion, in contrast to immunoreceptor tyrosine-based activation motifs (ITAMs) in the activating receptors [11,22]. The ITIM motif consists of six amino acids (S/I/V/LxYxxI/V/L). The classical ITIM has a modified version, which is the immunoreceptor tyrosine-based switch motif (ITSM) [23]. In ITSM, the first amino acid of the consensus sequence of the ITIM is replaced by a threonine (TxYxxI/V) [23]. Broadly speaking, inhibitory receptors can be classified into two groups: the immunoglobulin superfamily (IgSR) receptors and the C-type lectin inhibitory receptors [24]. A single V-type Ig-like domain in the extracellular region is the main feature of the IgSR receptors. Various IgSR receptors are expressed on myeloid cells, such as Fcγ receptors IIB (FcγRIIB) [25]; leukocyte immunoglobulin-like receptors/immunoglobulin-like transcript (LILRs/ILTs) [26]; leukocyte-associated immunoglobulin-like receptor (LAIR) [27]; signal regulatory protein α (SIRPα) [28]; CD300 glycoproteins CD300a and CD300f [29]; and sialic acid binding Ig-like lectins (Siglecs) [30]. The C-type lectin inhibitory receptors are named due to a calcium-dependent carbohydrate-binding protein motif in their structure, even though no apparent calcium binding or carbohydrate specificity is characterized in many C-type lectin inhibitory receptors [31]. Myeloid cells express various C-type lectin inhibitory receptors such as dendritic cell inhibitory receptor (DCIR) on dendritic cells (DCs) and mast cell function-associated antigen (MAFA) on mast cells, etc. [32,33].

## 3. Inhibitory Receptor Signaling

Inhibitory signaling pathways work coordinately with activating signaling pathways to initiate, terminate as well as modulate the intensity of the immune responses during infections. The appropriate signaling tone is achieved by balancing the contribution of both the activating receptors and inhibitory receptors. Activating receptors recruit ITAM-containing adaptor molecules and then ITAMs are phosphorylated by Src family kinases. This step creates docking sites for Src homology 2 (SH2) domain-containing kinases such as spleen tyrosine kinase (Syk) and zeta chain-associated protein kinase 70 (ZAP-70) [21]. Downstream signaling is initiated by the phosphorylation of the substrates of these kinases and activates immune responses. The inhibitory signal transduction initiates from the phosphorylation of ITIMs by Src family kinase in the cytoplasmic region upon inhibitory receptors binding to their ligands, resulting in a docking site to recruit phosphatase containing Src homology 2 (SH2)-domain, such as SH2-containing phosphatase 1 (SHP-1), SHP-2 and SH2-containing inositol phosphatase (SHIP-1) [24]. These phosphatases dephosphorylate downstream substrates to inhibit immune responses or maintain immune tolerance (Figure 1). The majority of inhibitory receptors recruit SHP-1 or SHP-2, except for FcγRIIB, which primarily recruits SHIP-1 [34]. Recruitment and binding of these phosphatases to the intracellular ITIM is specific and governs the extent of inhibition [35]. Binding of SHP-1 may lead to a distinctive inhibitory outcome compared with SHP-2. SHP-1 is active only when bound to ITIMs, while SHP-2 does not necessarily need a direct binding [36,37]. The traditional view that ITIM-mediated inhibitory receptor signaling opposes ITAM-mediated activating signaling is described above; however, there are exceptions to this rule. Some ITIM-bearing inhibitory receptors such as SIRP-α and LILRB1 can bind to c-Src tyrosine kinase (Csk) and lead to consequent SHP-independent inhibition [38,39]. Downstream of kinase (Dok) family members also mediates inhibitory signaling through Dok binding to Ras GTPase activating protein (Ras GAP) independent of direct binding to SHPs/SHIP [40]. Moreover, inhibitory receptors also regulate inhibitory signaling towards receptors that do not contain ITAMs, indicating a broader inhibitory capacity of these receptors [41,42]. In contrast to ITIM, ITSM can recruit adapter molecules of both activating and inhibitory signals, thus enabling a switch between activating and inhibitory signaling [43].

## 4. Viral Interaction with Inhibitory Receptors on Myeloid Cells

Inhibitory receptors on myeloid cells play an important role in regulating myeloid cell function to maintain immune homeostasis, which is essential for antiviral immunity [24]. However, during the co-evolution of viruses with their host, they acquired strategies to manipulate the host immune system to benefit their own replication. Increasing evidence reveals that viruses utilize inhibitory receptors on myeloid cells to facilitate virus entry, survival and replication, as well as escape from myeloid cell-mediated immune responses, ultimately enhancing viral pathogenesis [21]. Recent studies are reviewed here to illustrate how viruses exploit myeloid cell inhibitory receptors to enhance their own infections as well as evade surveillance and clearance using both innate and adaptive immune responses (Table 1).

### 4.1. Viral Manipulation of Inhibitory Receptors on Myeloid Cells to Enhance Infection

Viral entry into host cells is initiated by attachment to receptors followed by penetration through the cell membrane or direct cell membrane fusion [44]. Afterward, the viral genomes are transferred into host cells. Some inhibitory receptors, expressed on the cell surface, can serve as attachment factors and thus enhance viral entry into myeloid cells [45,46,47,48]. It seems that this process does not always trigger the inhibitory signaling pathway. 

CD300a, expressed on a macrophage surface, binds directly to Dengue virus (DENV) particles and enhances DENV internalization through clathrin-mediated endocytosis [49]. CD300a recognizes phosphatidylethanolamine (PtdEth) and phosphatidylserine (PtdSer) associated with virions and enhances viral entry. DCIR binds to a broad spectrum of sugars of both endogenous and exogenous origins [50]. Interestingly, Gp120-covered HIV-1 particles can act as multivalent ligands of DCIR. DCIR on immature DCs captures HIV-1 and promotes infection of CD4^+^ T cells [51,52]. Siglecs bind to sialoglycans [53], thus a variety of sialoglycans on viral surface proteins can also function as ligands of Siglecs. For example, HIV-1 membrane gangliosides can be recognized by a variety of inhibitory Siglecs on macrophages, including Siglec-3, Siglec-5, Siglec-7 and Siglec-9, and promote infection [54].

In addition to serving as attachment factors, upon binding to viral particles these inhibitory receptors sometimes trigger downstream inhibitory signaling pathways, thus contributing to myeloid cell-mediated immune modulation, which will be discussed next.

### 4.2. Viral Manipulation of Inhibitory Receptors to Evade Myeloid Cell-Mediated Antiviral Immunity

Immune evasion during viral infections is a long-discussed topic. The mechanism of immune evasion remains elusive and seems to be interconnected between innate and adaptive immunity. Myeloid cells, as an immune communication hub which bridges innate and adaptive immune responses, play a significant role in viral immune evasion. 

#### 4.2.1. Myeloid Cell-Mediated Viral Immune Evasion

In chronic viral infections, immune evasion is extensively studied, including both innate and adaptive immune escape. Viruses can evade host innate immunity through active disruption of antiviral responses. This is evident in a variety of chronic viral infections including Hepatitis C Virus (HCV) [55], Hepatitis B Virus (HBV) [56], Human immunodeficiency virus (HIV) [57] and Cytomegalovirus (CMV) [58], etc. Diverse mechanisms are involved through functional interactions between viral antigens, pathogen recognition receptors (PRRs), immune cells and signaling molecules in multiple myeloid cell types. 

Myeloid derived suppressor cells (MDSC) with significant immunosuppressive effects on both innate and adaptive immune responses attracted much attention recently [59]. MDSC uses multiple effector molecules and signaling pathways to regulate immune suppression [60]. Many chronic viral infections generate MDSC to induce viral persistence and promote T cell exhaustion. It was reported that MDSC-induced immunosuppression of T cell responses facilitates and maintains HCV persistence [61]. Polymorphonuclear MDSC (PMN-MDSC) was shown to maintain immune tolerance in replicating HBV and ameliorate hepatic tissue damage in chronic HBV infection [62]. A high frequency of MDSC was reported to correlate positively with viral loads and negatively with CD4^+^ T cell numbers in chronic progressive HIV-1 infection [63]. 

Although much research focuses on immune tolerogenic status in chronic infections, emerging evidence indicates that acute infections also adopt different strategies to evade immune surveillance and induce immune tolerance. These strategies include escaping PRRs recognition, inactivation of transcriptional factors, regulating the transcription and translation factors of innate immunity and antagonizing IFN pathways, etc. [64]. T cell exhaustion is not unique to chronic infections; in addition to evasion from innate immunity, it can also rapidly develop in acute viral respiratory infections [65] and acute neurological infections [66]. MDSC subsets were also shown to exacerbate certain acute infections including SARS-CoV2 [67,68,69].

Studies regarding the role of myeloid cells in virus-induced innate and adaptive immune evasion frequently focus on cytokines released by innate immune cells as well as viral interplay with activating pathways [1]. Triggering inhibitory receptor signaling on myeloid cells to induce innate and adaptive immune tolerance during viral infections is underestimated and not fully characterized. Emerging evidence shows that the expression of inhibitory receptors on myeloid cells is upregulated during both acute and chronic viral infections, indicating that these receptors may play important roles and should be studied intensively [70,71,72].

#### 4.2.2. Myeloid Cell Immune Checkpoints

The immune checkpoints of T cells and NK cells are identified and well characterized, and many of these checkpoints are already being developed as targets in cancer treatments [73,74]. Here, we highlighted several myeloid cell immune checkpoints reported to regulate myeloid cell-mediated immune responses and summarized how viruses manipulate these inhibitory receptors to evade antiviral immunity. We hope that this review will raise researchers’ interest in this emerging field and encourage researchers to explore the potential of myeloid cell immune checkpoints as new targets for antiviral immunotherapies.

##### LILRB

Inhibitory LILRB receptors were first identified in 1997 [75]. Their expression is enriched on myeloid cells as well as other immune cell types such as NK cells and T and B lymphocytes [76]. They are also expressed on additional cell types such as osteoclasts [77], leukemia [78], stromal and endothelial cells [79,80]. Endogenous ligands for LILRB receptors were identified, including human leukocyte antigen (HLA) class I molecules, S100A8/A9, some myelin-associated proteins and angiopoietin-like (ANGPTL) proteins [81]. Accumulating evidence proposes that LILRB receptors serve as myeloid cell immune checkpoints to control and limit overt immune responses [81]. Their expression is increased in suppressive macrophages and tolerogenic dendritic cells [82,83,84,85]. On macrophages, ligation of LILRB1 limits their phagocytic capability [86]. On monocytes, LILRB1 and LILRB2 agonism results in SHP-1 activation and downstream inhibitory signaling events to inhibit Ca^2+^ flux in response to Fc receptor engagement [87]. LILRB1, 2 or 4 activation on DCs maintains DCs in a tolerogenic state with impaired antigen presentation capability, leading to inhibition of T cell responses [88,89]. Although mice do not express LILRBs, they have the orthologous system with inhibitory PIR-B [90]. PIR-B negatively regulates antigen presenting activity of DCs and influences priming of cytotoxic T lymphocytes [91]. It was also reported to regulate differentiation of MDSC and facilitate tumor progression [92]. The association between LILRB receptors and tumors was extensively studied [78,81,93]; expression levels correlate with tumor growth and poor patient outcomes. Although not studied extensively to date, viruses also utilize LILRB receptors on myeloid cells to evade host immune surveillance [46]. During infection, viral proteins are degraded into small peptides and loaded onto HLA molecules. These peptide-HLA class I complexes can create ligands of LILRB receptors on macrophages, monocytes and DCs [94]. The most extensively investigated example is LILRBs’ role in HIV infection. The immunomodulatory function of the HLA-I/LILRB2 axis balances the immune response and correlates with a clinical HIV infection outcome [95]. A compromised DC function was found in HIV-infected patients as a result of interaction between LILRB2 and the HLA-B*35-Px allele [96]. In addition, LILRB2 binding to soluble HLA class I in HIV-1-positive plasma inhibited the allostimulatory functions of DCs [97]. The binding affinity to LILRB2 shows epitope specificity. For instance, a variant of HLA-B2705-restricted HIV-1 cytotoxic T cell epitope KK10 which creates ligands for LILRB2 increases receptor binding and leads to enhanced functional inhibition of DCs, thereby contributing to a tolerogenic status of myeloid cells [94]. Moreover, soluble HLA-G was significantly upregulated and interacted with LILRB2 on DCs during HIV infection [98,99]. This interaction initiated the inhibitory signaling pathway and led to functional impairment of DCs. This impairment of DCs ultimately results in improper adaptive immune responses and disease progression. In HIV patients, elevated serum interleukin-10 (IL-10) caused upregulated LILRB2 in monocytes which correlates with a dampened antigen-presenting ability of myeloid cells over-expressing LILRB2 [70]. In addition to HIV, other viruses also utilize LILRB receptors to facilitate immune evasion. A well-known example is antibody-dependent enhancement (ADE) which is postulated to be the reason for severe Dengue secondary infection [100,101]. Antibody-opsonized DENV can directly bind to LILRB1; however, this co-ligation does not enhance virus entry. Instead, this co-ligation is utilized by DENV to inhibit activating Fcγ receptor (FcγR) signaling, which results in SHP-1 recruitment, dephosphorylation of Syk and down-regulation of the IFN-stimulated gene (ISG) expression in monocytes. Triggering the inhibitory signaling pathway facilitates the evasion of early antiviral response and augments secondary DENV infection (Figure 2A) [100].

##### DCIR

DCIR is expressed on DCs and macrophages, as well as other immune cell types including monocytes, granulocytes and B lymphocytes [47]. It contains an ITIM which mediates inhibitory signals after ligation to the ligands [102]. As an inhibitory receptor on DC cells, DCIR triggering inhibits Toll-like receptor 8 (TLR8)-dependent IL-12 and tumor necrosis factor-α (TNFα) production via activation of the intracellular ITIM motif, whereas TLR2, TLR3 and TLR4-induced cytokine levels are not affected [103]. Expressed on macrophages, upon activation, DCIR inhibits CpG-ODN-induced expression of pro-inflammatory cytokines [104]. Although DCIR’s involvement in virus entry and transmission was studied extensively, little research was performed to elucidate the role of the DCIR-triggered inhibitory signaling pathway in virus-induced immune responses. It was reported that SHP-1, SHP-2, Syk and Src kinases, as well as protein kinase C-alpha (PKC-α) and mitogen-activated protein (MAP) kinases, are all involved in the DCIR-mediated signaling pathway triggered by HIV-1 [105]. The tyrosine and threonine residues of ITIM motif were shown to play a pivotal role in this HIV-1-mediated signaling which indicates that the inhibitory signaling pathway through ITIM motif may be triggered [105]. Blocking HIV-1 binding to DCIR on DCs significantly decreases exosome release [106]. These secreted extracellular vesicles (EVs) by HIV-pulsed DCs contain pro-apoptotic protein DAP-3 and induce apoptosis in uninfected CD4^+^ T lymphocytes (Figure 2B). These findings indicate that the co-ligation of HIV-1 with DCIR can weaken the specific immune responses to viral infection, and therefore increase immune tolerance. It is noteworthy that DCIR is also involved in type I IFN responses [50]. In mice, DCIR deficiency impairs STAT-mediated type I IFN signaling in DCs, leading to increased IL-12 and differentiation of T lymphocytes toward type 1 T helper (Th1) cells during tuberculosis infection. As a consequence, DCIR-deficient mice control *Mycobacterium tuberculosis* better than wild-type (WT) animals. It was also reported that in some DC subsets DCIR inhibits IFN responses [107]. Whether the inhibition or sustaining effect of type I IFN responses contributes to viral immune evasion is still elusive and needs further investigation.

##### Siglecs (Inhibitory)

Siglecs are cell-surface receptors belonging to the immunoglobulin superfamily. They comprise 2–17 extracellular Ig domains and ITIM-containing cytoplasmic domains for most Siglecs, including Siglec-2, Siglec-3 and Siglec-5–12 [30]. Several Siglecs, such as Siglec-14, Siglec-15 and Siglec-16, contain a positively charged residue in their transmembrane domain to complex with ITAM-containing adaptor proteins and are predicted to be activating receptors [108]. Siglec-1 does not contain cytosolic signaling motif. Most human Siglecs appear in pairs (activating receptors and inhibitory receptors) with similar extracellular regions but divergent transmembrane and cytoplasmic regions [109]. Here we focus on discussing inhibitory Siglecs and their role as myeloid cell immune checkpoints in virus-induced immune evasion and tolerance. Siglec-G was identified as the mouse homolog of human Siglec-10 [110] and its role in viral evasion of innate immunity was demonstrated using mouse experiments [71]. RNA viruses such as Vesicular stomatitis virus (VSV) and Sendai virus (SeV) infection upregulate *Siglecg* expression in macrophages, leading to degradation of retinoic acid-inducible gene I (RIG-I) and suppression of the interferon-β (IFN-β) response (Figure 2C). *Siglecg^−/−^* macrophages produced more IFN-β, TNF-α and IL-6 protein when infected with VSV or SeV. VSV-infected *Siglecg*-deficient mice showed decreased viral titers and mortality which are also associated with increased IFN-β. The mechanism functions through a negative feedback: RNA virus infection upregulates *Siglecg* expression, then Siglec-G suppresses type I IFN production by promoting c-Cbl-mediated K48-linked ubiquitination and proteasomal degradation of RIG-I. This negative feedback explains how RNA viruses utilize the Siglec-G receptor-mediated inhibitory signaling pathway to evade innate immune surveillance [71]. A recent paper reported that α2,6-biantennary sialoglycans of HBV surface antigen (HBsAg) can bind human Siglec-3 in vitro by immunoprecipitation and ELISA [111]. This binding triggered a Siglec-3-mediated inhibitory signaling pathway on myeloid cells and suppressed TLR ligand-induced cytokine production. This induced immunosuppression can be reversed using an antagonistic anti-Siglec-3 mAb (10C8). Furthermore, a higher expression of Siglec-3 was associated with an increased risk of hepatocellular carcinoma (HCC) in chronic hepatitis B (CHB) patients, indicating that targeting Siglec-3 may not only reverse HBV-induced immune suppression, but also beneficially reduce HCC incidence in CHB patients. 

##### SIRPα

Signal regulatory protein α (SIRPα) is a myeloid cell immune checkpoint which is especially abundant in myeloid cells such as macrophages, whereas it is barely expressed in T cells, B cells, NK cells and natural killer T (NKT) cells [28]. SIRPα is a transmembrane protein with a cytosolic region that contains two ITIM motifs [28]. Upon binding to CD47, SIRPα induces downstream ITIM-mediated inhibitory signaling and negatively regulates phagocytosis of macrophages [112]. Although first identified as an inhibitory receptor in phagocytosis pathway, blockade of the CD47-SIRPα axis also leads to increased activation of antigen-presenting cells (APCs) which bridges innate and adaptive immune responses, thereby enhancing T and B cell-mediated immunity [113,114,115]. Indeed, the disruption of CD47-SIRPα signaling was demonstrated to lead to tumor regression through mechanisms such as promoting phagocytic uptake of tumor cells by macrophages and increasing antigen presentation by DCs, thereby stimulating anti-tumor adaptive immune responses [116]. Moreover, the blocking interaction of CD47 and SIRPα was also demonstrated to reprogram MDSC and tumor-associated macrophages (TAMs) as non-suppressive [117,118]. Quite a few anti-CD47 and anti-SIRPα antibodies are already undergoing clinical trials for tumor treatment [119,120]. There is growing evidence that disrupting the CD47 and SIRPα interaction may also benefit antiviral immunity during viral infections [121]. Viruses adopt different strategies to manipulate this “don’t eat me” signaling. Poxviruses encode a CD47 mimic which works as a virulence factor, while a variety of viruses including HCV infection and severe acute respiratory syndrome coronavirus 2 (SARS-CoV-2) upregulate CD47 expression in infected cells without encoding a ligand by themselves. Furthermore, proinflammatory cytokines released after viral infections can upregulate CD47 on uninfected DCs, thus bridging innate suppressive modulation with downstream adaptive immune responses [121]. Studies show that blocking the CD47-SIRPα axis releases the host from innate immune suppression and enhances both antiviral innate and adaptive immunity [122]. Blockade of the CD47-SIRPα axis by anti-CD47 antibody, MIAP410, in mice infected with lymphocytic choriomeningitis virus (LCMV) increased activation of macrophages and DCs as well as improved CD8^+^ T cell responses, probably by strengthening the APC capacity of DCs resulting in improved virus control (Figure 2D) [122,123]. Similarly, in HIV-1-infected humanized mice, blockade of CD47 reduced HIV antigen level and restored the number of CD4^+^ and CD8^+^ T cells to a level comparable to healthy uninfected mice [122]. The depletion of CD8^+^ T cells led to impaired virus control which means that the antiviral effect is mainly dependent on CD8^+^ T cells. Interestingly, the CD47-SIRPα axis seems not only involved in innate immunity and T cell responses, but also in antibody responses. CD47-deficient mice showed a higher level of IgG production and greater protective efficacy against a lethal challenge of influenza post-vaccination compared to wild-type mice [124]. Recent in vitro data showed that SARS-CoV-2 infection increases SIRPα levels on primary human monocytes [125]. In silico analysis of the expression pattern of immune inhibitory receptors during COVID-19 infection showed an increased expression of SIRPα, as well as the aforementioned LILRBs and Siglecs, in bronchoalveolar fluid macrophages in SARS-CoV-2 patients, although how the increased expression affects the SARS-CoV-2 viral infection needs further investigation [72]. 

**Table 1 viruses-14-01144-t001:** Viral manipulation of inhibitory receptors on myeloid cells.

Inhibitory Receptors	Myeloid Cell Expression *	Functional Importance	Reference
CD300a	MC, Mac, Mono, DC, N, E, B	Recognizes PtdEth and PtdSer associated with virions and enhance viral entry.	[49]
LILRB	Mac, Mono, N, DC, E, B	Antibody-opsonized DENV utilizes LILRB1 to inhibit FcγR signaling in monocytes and inhibit ISG expression.	[100,101]
		KK10 epitope variant of HIV-1 binds to LILRB2 leading to a tolerogenic phenotype of DCs.	[94]
		HIV elevates LILRB2 expression via IL-10 and inhibits APC capacity of monocytes, thus impairing function of CD4^+^ and CD8^+^ T cells.	[70,126]
DCIR	Mac, Mono, N, DC	HIV-1 utilizes DCIR on DC to infect CD4^+^ T cells and induces apoptosis in uninfected CD4^+^ T cells.	[51,52,106]
Siglecs	MC, Mac, Mono, N, DC, E, B	HIV-1 membrane gangliosides can be recognized by a variety of inhibitory Siglecs on macrophages including Siglec-3, Siglec-5, Siglec-7 and Siglec-9 and promote infection.	[54]
		VSV upregulates Siglec-G expression on macrophages, leading to RIG-I degradation and IFN-β response suppression.	[71]
		α2,6-biantennary sialoglycans of HBsAg binding to human Siglec-3 activates Siglec-3 on myeloid cells and induces immunosuppression.	[111]
SIRPα	Mac, N, DC	CD47-SIRPα blockade by anti-CD47 antibody increased activation of macrophages and DCs, improved CD8^+^ T cell responses by strengthening APC capacity of DCs, contributing to LCMV control.	[122,123]
		Blockade of CD47 reduced HIV antigen level and restored CD4^+^ and CD8^+^ T cell numbers.	[122]

* MC, mast cell; Mac, macrophage; Mono, monocyte; N, neutrophil; DC, dendritic cell; E, eosinophil; B, basophil.

### 4.3. Polymorphism of Inhibitory Receptors on Myeloid Cells Associates with Viral Susceptibility

As discussed above, viral manipulation of myeloid cell inhibitory receptors enhances viral infection as well as facilitates viral immune escape and tolerance. Thus, the polymorphism of these receptors is associated with the susceptibility and severity of viral infections; this was demonstrated in the literature. The association of LILRB1 polymorphism in the regulatory and ligand-binding regions with HCMV infection in transplant patients was found and reported [127]. Specific LILRB1 alleles enable superior immune evasion by HCMV via modulation of LILRB1 expression and interaction with its ligand HLA-G, leading to fetal tolerance. Genetic association studies also revealed that polymorphism of Siglecs among different mammals contributes to different susceptibilities to infectious diseases, degenerative disorders and cancers [20,128]. 

## 5. Inhibitory Receptors on Myeloid Cells as Future Therapeutic Target for Antiviral Treatment

Within the last decade, immune checkpoint inhibitors achieved great success through the introduction of T cell-targeted immunomodulators blocking the immune checkpoint targets CTLA-4 and PD1 or PDL1 [129,130]. Emerging evidence suggests that immune checkpoint inhibitors may also have promise for the treatment of viral infections [21,131,132]. Indeed, several immune checkpoint agents are currently in clinical trials aiming to treat different viruses, including SARS-CoV-2 (NCT04413838, NCT03407105, NCT04343144, NCT04638439 and NCT02028403). Notably, all of these agents are T cell-targeted. Alleviating virus-induced immunosuppression by targeting myeloid cells is still largely unexplored. There are a limited number of preclinical proof-of-concept studies which highlight the potential of this approach using tool antibodies. A good example is an anti-CD47 antibody blocking the CD47-SIRPα axis [121]. Several anti-CD47 antibodies, such as MIAP301, MIAP410, Hu5F9-G4, CC-90002, SRF231, B6H12.2 and ALX148, were developed and tested in cancer and infection treatment [119,120,122,133,134]. MIAP410 is an anti-mouse CD47 antibody and its treatment in virus-infected mice results in an increased proliferation and activation of innate and adaptive antiviral immunity [122].

Despite the supportive evidence for targeting myeloid cell inhibitory receptors as a novel antiviral strategy, several important questions remain. To date, most research about myeloid cell immune checkpoints focused on chronic viral infections [46,48,106,121]. Whether or not these findings regarding chronic infections are also applicable to acute viral infections needs to be established. Considering the significant role of myeloid cells in inducing antiviral innate immunity in the early acute infection phase, as well as the different mechanisms of immune evasion, suppression and tolerance between acute and chronic infections, targeting inhibitory receptors on myeloid cells may generate different outcomes in acute viral infections. Emerging literature suggests that the expression profile of these myeloid cell inhibitory receptors was altered in acute infections [72,135,136]. More research is needed to better understand the mechanism and function of these expression changes. Additionally, the antiviral approach of targeting the inhibitory receptors on myeloid cells relies on mediating the host immune response, which is a very different proposition to a direct antiviral mechanism. Therefore, we can expect that the outcome of preclinical studies will depend on appropriate animal models. Any species differences in the regulation of the host immune system may represent a significant translational challenge from pre-clinical experiments to human trials. Humanized mice or other species that have a similar expression profile of these targeted receptors can be considered. Finally, appropriate dosing regimens and combinations with standard-of-care or other immunomodulatory agents should be rationally designed to achieve therapeutic goals while avoiding over-activation of the host immune system.

## 6. Concluding Remarks

Numerous studies revealed clear contributions of inhibitory receptors to myeloid cell function as well as innate and adaptive immune responses. These inhibitory receptors show potential to be targeted as antiviral treatments. Since viruses, especially within the same family, usually share a conserved structure of proteins to interact with intracellular signaling, it is possible that viruses utilize similar inhibitory signaling pathways for escaping immune responses. Thus, targeting inhibitory receptors on myeloid cells presumably has a broad spectrum of applications for fighting against different viruses. Currently, there is a need to better understand the role of such receptors in immune responses as well as the interaction between viruses and these receptors. Given the burgeoning knowledge regarding myeloid cell inhibitory receptors and the myeloid cell checkpoint inhibitors developed in the cancer research field, we anticipate that these studies will also shed light on novel antiviral therapeutic approaches in the near future.

## Figures and Tables

**Figure 1 viruses-14-01144-f001:**
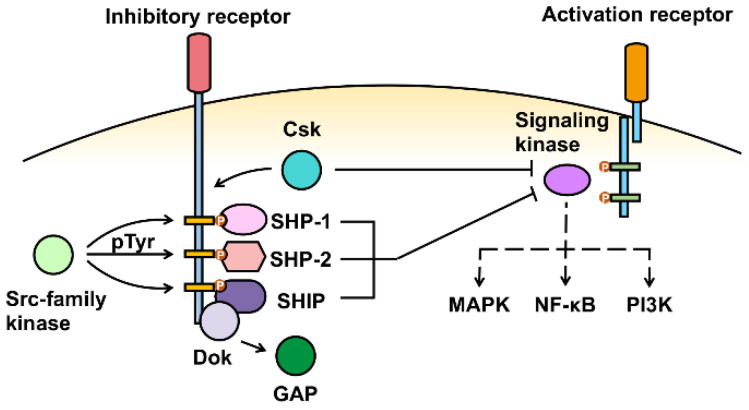
Inhibitory signaling. Inhibitory receptors recruit a Src-family kinase to phosphorylate intracellular immunoreceptor tyrosine-based inhibitory motif (ITIM) which creates docking sites for phosphatases SHPs/SHIP. The recruitment and binding of SHPs/SHIP to inhibitory receptors suppress proximal activation signals mediated by the signaling kinases recruited to the activation receptors. Alternatively, some inhibitory receptors can work through Csk or Dok-mediated inhibition independent of binding to SHPs/SHIP.

**Figure 2 viruses-14-01144-f002:**
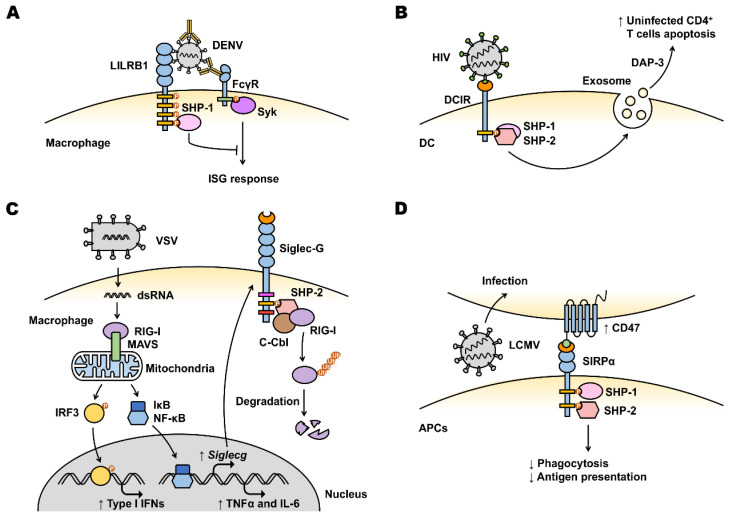
Viruses manipulate host inhibitory receptors for immune evasion. (**A**) Dengue viruses (DENV) opsonized with antibodies are taken up via Fcγ receptors (FcγR) on macrophages. By co-ligating LILRB1, DENV can inhibit FcγR signaling and IFN-stimulated gene (ISG) responses. (**B**) Human immunodeficiency virus (HIV) attaches to dendritic cells (DCs) via DCIR. HIV-pulsed DCs release more exosomes containing pro-apoptotic protein DAP-3, inducing apoptosis of neighboring uninfected CD4^+^ T lymphocytes. (**C**) RIG-I recognizes vesicular stomatitis virus (VSV) and induces *Siglecg* expression. The increased Siglec-G inhibits immune response by recruiting phosphatase SHP-2 and the E3 ubiquitin ligase c-Cbl to RIG-I, resulting in RIG-I degradation. (**D**) Cells infected with lymphocytic choriomeningitis virus (LCMV) have an upregulated expression of CD47, which mediates the “don’t eat me” signal by interacting with SIRPα. The CD47-SIRPα axis impairs phagocytosis and antigen-presenting capabilities in antigen-presenting cells (APCs).

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
