# Peer review of "New Targets for Antiviral Therapy: Inhibitory Receptors and Immune Checkpoints on Myeloid Cells"

_viruses, 2022, doi:10.3390/v14061144_

Round 1
Reviewer 1 Report
The authors have answered my previous concerns and the manuscript is now ready for acceptance.
Reviewer 2 Report
After revision, the manuscript was well organized and can be accepted for publication in this journal.
Reviewer 3 Report
The authors have properly addressed the questions and comments being raised in the earlier review of the article. I recommend the paper to be accepted for publication.
This manuscript is a resubmission of an earlier submission. The following is a list of the peer review reports and author responses from that submission.
Round 1
Reviewer 1 Report
The manuscript by Lui and He, describes the role of inhibitory receptors (IR) expressed by myeloid cells and their manipulation by viruses during infection, to evade the immune response, and points to increasing interest in targeting these receptors to treat viral infections.
The growing consideration of these IR in cancer and infection is well illustrated by the many recent publications and good reviews, which are cited by the authors. However, the manuscript, which is presented as a summary, suffers from a lack of clarity and the main questions are treated superficially.
After a brief description of some characteristics of IR on myeloid cells, the authors provide a list of four types of IR with examples of viral interplay. A short section is then dedicated to the announced subject of the review about “New therapeutic targets for antiviral therapy”.
Throughout the manuscript, there is a lack of precision and missing information, making it difficult to understand the structure and function of the IR in healthy conditions (homeostasis) and in which way viruses interact and modify their function. While IRs can be considered as potential therapeutic targets, there are many points worth discussing critically, as expected in a review, for example:
- complexity of the IR, their level of expression in different situation, their ligands. (direct interaction of the virus or induced ligand upon infection)
- will it be the same considering persistent/chronic infection or acute infection?
- pertinence of animal model expressing IR and IFN signaling molecules, which differed from the human ones.
- IR are expressed on different cells: is it possible to ensure that their inhibition will not alter other functions, considering the network of interaction between signaling pathways required to maintain homeostasis and restore it after viral infection.
In the concluding remarks, the authors hypothesized that it could be “possible that viruses utilize similar inhibitory signaling for escaping immune response”. However, it should be discussed that viruses in a same family can develop different strategies to counteract the innate immune response.
To be added, the writing is often inappropriate and many sentences have syntax errors. For example:
-lane 25: “to be paid more attention to due to”
-lane 35: “people focus more” (researchers?)
-lane 44: “inhibitors have been under development” (are under development, or have been developed?)
-Lane 46: ‘viruses…manipulate inhibitory receptors to down regulate antiviral responses for enhanced viral pathogenesis” (meaning unclear)
-lane 52: IRs “recruit adaptor molecules containing ITIM located in the cytoplasmic portion” (does not reflect IR structure)
-lane 99: “LILRB receptors…. they are also reported to express on osteoclasts”
-lane 209: “All the evidence offers the hope of targeting inhibitory receptors…” difficult to understand . in fact, the writing of the whole paragraph 5 is difficult to read
Reviewer 2 Report
Liu et al. discussed the structure and signaling of inhibitory receptors on myeloid cells and how viruses interact with these receptors to play a role in virus-host interactions. In this review, four types of inhibitory receptors were highlighted, and how viruses exploit inhibitory receptors to evade surveillance and clearance by both innate and adaptive immune responses was illustrated. The status of utilizing inhibitory receptors on myeloid cells as therapeutic targets for antiviral treatment was analyzed. These inhibitory receptors showed potential to be targeted as antiviral treatments. Some minor revisions are as follows:
- Many inhibitory receptors have specific intracellular sequence motifs, like immunoreceptor tyrosine-based inhibitory motifs (ITIMs). However,the function of ITIMs is not clearly summarized.
- It seems that different subsets of myeloid cells use SHP-1 to regulate different signaling pathways. The summary of the inhibitory receptor signal is not comprehensive enough.
- The authors summarized four inhibitory receptors that interact with viruses on myeloid cells. There are other inhibitory receptors that are also expressed on myeloid cells, such as LAIR1, TIM3. Why did the authors choose these inhibitory receptors?
- Do myeloid cells have any special advantages in antiviral treatment? It should better be briefly introduced.
Reviewer 3 Report
The immune response is mediated by intricate positive and negative regulatory signals. In the previously, people focus more on the pathways activating immune cells. However, over the last two decades, it has become increasingly apparent that inhibitory signals can regulate various immune cell checkpoints. In this review, authors mainly focus on the inhibitory receptors on myeloid cells. Basic structure and mechanism of inhibitory receptors were discussed. Finally, utilizing inhibitory receptors as pharmacological targets for immune suppression were presented and discussed. The paper has certain advantages for this field research work, and has value for publishing in journal of Viruses. I suggest this manuscript can be published after the following minor revisions:
- The inhibitory receptor signaling: Since inhibitory receptors binding SH2 containing phosphatases (SHP-1, SHP-2 and SHIP-1), is there a differential role for SHP-1 and SHP-2 mediated inhibition?
2. The inhibitory signal needs to bind its ligand phosphatases, is the inhibitory pathway dependent of phosphatase activity?
Round 2
Reviewer 1 Report
As mentioned in my previous report the manuscript by Liu and He suffers from a lack of clarity all over the text. By adding only a few sentences to the new version, the authors did not sufficiently improve their manuscript. Moreover, only the few examples given of errors in the text have been partially corrected and therefore, the quality of the writing has not been improved either and would require English editing.
The authors have not yet provided the reader with enough detail for an easy understanding of the subject covered in their review.
Indeed, regarding the responses provided by the authors to the various comments that have been raised, it is clear that the targeting of IRs expressed on myeloid cells to treat an acute viral infection is still speculative. The question is of interest, but as previously mentioned it would be important to better explain it in the manuscript and discuss it critically. Although again relating to chronic infection, the recent paper: SIGLEC-3 (CD33) serves as an immune checkpoint receptor for HBV infection: Tsai et al J Clin Invest (2021), should be mentioned.
